# Deformation Characteristics and Constitutive Equations for the Semi-Solid Isothermal Compression of Cold Radial Forged 6063 Aluminium Alloy

**DOI:** 10.3390/ma14010194

**Published:** 2021-01-03

**Authors:** Yongfei Wang, Shengdun Zhao, Yi Guo, Kuanxin Liu, Shunqi Zheng

**Affiliations:** 1School of Mechanical Engineering, Xi’an Jiaotong University, Xi’an 710049, China; wangyongfei324@mail.xjtu.edu.cn (Y.W.); sdzhao@mail.xjtu.edu.cn (S.Z.); 2State Key Laboratory of Materials Processing and Die & Mould Technology, Huazhong University of Science and Technology, Wuhan 430074, China; 3School of Energy and Power Engineering, Xi’an Jiaotong University, Xi’an 710049, China; 4Ningbo Branch of China Ordance Academy, Ningbo 315103, China; liukuanxin@cmari.com (K.L.); 2010010024@mail.hfut.edu.cn (S.Z.)

**Keywords:** semi-solid isothermal compression, aluminum alloy, microstructure, deformation characteristics, deformation mechanism, constitutive equations

## Abstract

Al-Mg-Si based alloys are popular alloys used in the automotive industry. However, limited studies have been performed to investigate the microstructure, deformation characteristics, and deformation mechanism for the semi-solid 6063 alloys. In this study, the cold radial forging method and semi-solid isothermal treatment (SSIT) are proposed in the semi-solid isothermal compression (SSIC) process to fabricate high-quality semi-solid 6063 billets. The effects of deformation temperature, strain rate, and strain on the microstructure, deformation characteristics, and deformation mechanism of the SSIC of cold radial forged 6063 alloys were investigated experimentally. Constitutive equations were established based on the measured data in experiments to predict the flow stress. Results show that an average grain size in the range from 59.22 to 73.02 μm and an average shape factor in the range from 071 to 078 can be obtained in the microstructure after the cold radial forged 6063 alloys were treated with SSIT process. Four stages (i.e., sharp increase, decrease, steady state, and slow increase) were observed in the true stress- true strain curve. The correlation coefficient of the constitutive equation was obtained as 0.9796 while the average relative error was 5.01%. The deformation mechanism for SSIC of cold radial forged aluminum alloy 6063 mainly included four modes: The liquid phase flow, grain slide or grain rotation along with the liquid film, slide among solid grains, and the plastic deformation of solid grains.

## 1. Introduction

Al-Mg-Si based aluminium alloys are popular materials applied in various engineering applications due to their outstanding physical properties, which are usually used to fabricate products via hot forging and machining processes. However, some drawbacks are observed during the hot forging manufacturing process, which includes low efficiency, high production cost, and low material usage coefficient [1,2,3]. The high-pressure die casting process has also been applied for the fabrication of aluminium alloy components, but the turbulent flow in this process causes the gas and shrinkage porosity [4], due to the high filling rate [5].

These disadvantages including low efficiency, high production cost, and low material usage coefficient, gas and shrinkage porosity, can be effectively eliminated by the semi-solid metal forming (SSMF) process, which integrates the easy fluidity of liquid casting process and the high mechanical properties of solid-state plastic forming and is considered one of the most promising metal processing technologies in the 21st century [6,7,8]. In the SSMF process, the semi-solid billet, which has small- and near-spherical microstructures, is used to fabricate high-quality components with a dense internal structure, accurate size, few macro defects, and few micro defects [9].

In the SSMF process, the deformation characteristics and mechanism of the semi-solid billets are vital for the prediction of the flow stress, based on the temperature, material structure, and strain rate, through which the optimisation of the process parameters can be achieved [10,11]. Therefore, a lot of research has been performed to study the deformation characteristics and mechanisms in recent decades of various aluminium alloys [11,12,13,14,15,16]. Xu et al. [7] investigated the compression behaviour of the semi-solid AZ91D magnesium alloy. The semi-solid billets were prepared by the strain-induced melt activation (SIMA) process for the isothermal compression test by employing the repetitive upsetting-extrusion (RUE) method. Three stages were observed in experiments in the compression process. The predominant mechanism in each of these three stages was identified through the microstructure analysis. Chen et al. [11] discussed the deformation characteristics of the extruded 7075 aluminum alloy processed in the temperature range from 350 to 600 °C. It was found that the constitutive characteristics in the studied temperature range up the semi-solid temperature could be classified as plastic (350 to 490 °C) and thixotropic (above 490 °C) deformations. Binesh and Aghaie-Khafri [12] reported the mechanical properties of the 7075 aluminum alloy processed by the semi-solid SIMA. The semi-solid billet was prepared by applying the compression at ambient temperature and then at 600–620 °C (semi-solid temperature) for 5–10 min. It was revealed that the sample processed by SIMA at 600 °C for 10 min exhibited the largest values of both peak and steady-state stresses during the compression test. Wang et al. [13] studied the deformation characteristic of semi-solid ZCuSn10 copper alloy during the isothermal compression. The semi-solid billet used in this work was fabricated by the SIMA method including the rolling and remelting process. It was concluded that the resistance to deformation for the ZCuSn10 alloy under the semi-solid condition decreased with the increase in the process temperature. These studies indicate that the deformation characteristics of semi-solid materials are of importance for optimizing the semi-solid isothermal compression (SSIC) process.

The preparation of the semisolid billet is an indispensable step for the SSIC process. The SIMA, as one of the most popularly studied process methods for the preparation of the billet with fine and spheroidal solid particles, includes two stages namely the plastic deformation and the semi-solid isothermal treatment (SSIT). In the SIMA process, plastic deformation is the most critical stage as it significantly influences the grain size and grain distribution uniformity in the microstructure of the semi-solid alloy. Therefore, different deformation methods applied in the SIMA process, such as compression [12,17], rolling [13], repetitive upsetting-extrusion [18], equal channel angular extrusion [19,20], has attracted the attention of researchers in recent decades. Compared to the traditional deformation methods used in the SIMA process, several benefits to the industrial application can be obtained by adopting the cold radial forging technique in the SIMA process, such as large semi-solid billet, high level of automation, and easy operation [21,22,23].

However, to the best knowledge of the authors, the aluminium alloy directly treated by cold radial forging in the SSIC process has not been reported. Moreover, little research has been performed on the microstructure, deformation characteristics, and deformation mechanism for the semi-solid 6063 alloys. In this study, the cold radial forging method is proposed in the SIMA process for the preparation of high-quality semi-solid 6063 billets for the SSIC process. The effects of deformation temperature (i.e., isothermal temperature), strain rate, and strain on the microstructure, deformation characteristics, and deformation mechanism of the cold radial forged 6063 alloys were investigated experimentally. Furthermore, the constitutive equation of the semi-solid 6063 alloys was established based on the experimental results to predict the flow stress based on the effective liquid phase fraction, the deformation temperature, strain rate, and strain.

## 2. Materials and Experimental Procedure

The chemical composition of 6063 aluminium alloy investigated in this study is summarized in Table 1. The solidus and liquidus temperatures of this alloy were identified by the heat flow-temperature curve, which was obtained by using an ‘STA 449F5’ differential scanning calorimeter (DSC) (Netzsch, Selb, Germany). The samples with the dimension of 6 mm × 6 mm × 5 mm were heated from room temperature to 750 °C at 10 °C/min under a nitrogen atmosphere and analyzed by DSC. The results show that the solidus and liquidus temperatures of this alloy are 615 °C, and 655 °C, respectively. The largest weight proportion of the chemical composition is Mg, which accounts for 0.51% and is followed by Si and Mn. Ti and Cr are less than 0.001%, and 0.005% in weight percentage, respectively.

The SSIC process of the cold radial forged 6063 aluminium alloy in this paper mainly includes three stages as shown in Figure 1. Stage I: The barstock of 6063 alloys with a diameter of 100 mm was deformed by the cold radial forging process until the reduction in the area reached 64% (i.e., *ϕ*60 mm). The reduction in the area is calculated by the ratio of (A0–A1) to A0, where A0 and A1 are the cross-sectional area of the initial alloy, and the deformed alloy, respectively. Then, the action of sampling was finished from the cold radial forged alloy along the direction of radial forging, which had the diameter and height of 10 mm and 15 mm, respectively. Stage II: The SSIC process including SSIT and the deformation of isothermal compression were carried out with the SSIC- equipment which mainly included the top pressing head, bottom pressing head, insulation material, resistance furnace, and thermocouples. During the SSIC process, the resistance furnace was heated up to the setting deformation temperature, and then the cold radial forged sample was placed between the top and bottom pressing head for SSIT. The semi-solid samples can be obtained when the SSIT process had been executed for 10 min at different isothermal temperatures. After that, these samples were compressed with the true strain of 0.7 to obtain the compressed semi-solid sample by the top and bottom pressing head at different deformation temperatures and different strain rates. For all the experiments, the deformation temperature was selected the same as the isothermal temperature in SSIT. Stage III: The compressed semi-solid sample was cooled rapidly by water quenching for preserving the microstructures of the sample.

In this paper, there was a total of 12 semi-solid isothermal compression experiments performed to examine the effects of the operating parameters on the microstructure, deformation characteristics, and the deformation mechanism of the cold radial forged 6063 aluminum alloy. The details of the experiments are summarized in Table 2. The operating parameters studied in this paper contained the isothermal temperature, the strain rate, and the strain in Stage II. The isothermal holding time in all experiments was set as 10 min. In Experiments 1–3, the isothermal temperatures varied from 625 to 635 °C with no deformation supplied. The strains in Experiments 4–12 were all set as 0.7. In Experiments 4–6, the isothermal temperatures were all selected as 625 °C at the strain rate of 0.01, 0.1, and 1 s^−1^, respectively. Similarly, in Experiments 7–9 and Experiments 10–12, the isothermal temperatures were the same in each group with the strain rate varying from 0.01 to 1 s^−1^.

During the semi-solid isothermal compression experiments, the true stress and true strain are determined in-situ by the SSIC-equipment (Xi’an Jiaotong University, Xi’an, China). Samples were taken along the axial direction, first ground through the sandpapers with the grits of 200, 400, 600, 800, 1000, 1200, and 1500 in sequence and then polished through the diamond paste with the particle size of 0.1 μm. Samples were etched in the aqueous HF solution with a concentration of 5% for about 90 s at room temperature. Finally, the observation of sample microstructure was carried out by optical microscope (NIKON ECLIPSE LV 150N, Nikoon, Tokyo, Japan). The average grain size (*D*) of solid grains can be calculated by Equation (1) while the shape factor (*F*) can be obtained by Equation (2) based on the number, area, and perimeter of solid grains, which were measured and calculated by the Image-Pro Plus 6.0 software (Media Cybernetics, Rockville, MD, USA) [24,25,26,27]. The effective liquid phase fraction in semi-solid samples $f_{EL}$ was calculated by Equation (3) [27],
(1)$$D=\frac{\sum_{N=1}^{N}\sqrt{4A/\pi }}{N}$$
(2)$$F=\frac{\sum_{N=1}^{N}4\pi A/P^{2}}{N}$$
(3)$$f_{EL}=\frac{A_{L}}{A_{L}+A_{S}}$$
where *A*, *N*, *P* are the area, number, perimeter of solid grains, respectively; *A*_L_ and *A*_S_ are the area of the liquid phase, and the area of the solid phase, respectively.

## 3. Results and Discussion

### 3.1. Effects of SSIT on the Microstructure

The microstructures of the radially forged 6063 aluminum alloy without and with SSIT are shown in Figure 2. It can be seen from Figure 2a that an obvious texture can be observed in the microstructure. The grains were elongated along the radial forging direction. Figure 2b–d show the semi-solid microstructures obtained after the radial forged alloy processed by SSIT at the isothermal temperature of 625 °C, 630 °C, and 635 °C for 10 min, respectively. Under the isothermal temperature condition of 625 °C, the semi-solid material of 6063 aluminum alloy was prepared with the fine, nearly spheroidized, uniform solid grains, but the liquid film between the solid grains in the microstructure was relatively thin, which can be seen in Figure 2b. Comparing Figure 2c,d, the solid grains gradually grew and nearly spheroidized, and a thicker liquid film was observed separating these solid grains as the isothermal temperature increased from 630 to 635 °C.

The characteristics of the microstructure can be obtained by the quantitative analysis of samples processed at different isothermal temperatures, which includes the average grain size, the shape factor of solid grains, and the effective liquid fraction. As shown in Table 3, it was found that the average grain size, shape factor, and effective liquid fraction of solid grains increased with the enlargement in the isothermal temperature. During the SSIT process, the protruding edges and corners of solid grains will dissolve and subsequently precipitate at the sunken regions due to the difference in curvatures of different parts of the single solid grain [21]. Therefore, the shape factor further improves when the isothermal temperature increases from 630 °C to 635 °C. The largest average grain size and the smallest average shape factor were found as 73.02 µm and 0.71 µm, respectively. The maximum and minimum effective liquid fractions were obtained as 11.10%, and 14.42%, respectively. This means that the semi-solid material obtained from the radial forged 6063 aluminium alloy applied in this study after the SSIT is qualified for the semi-solid forming as the semi-solid materials with microstructure characteristics with the average grain size blow 100 μm and a shape factor above 0.6 is suitable for the semi-solid forming [28,29].

### 3.2. True Stress-True Strain Curves and the Deformation Pattern

The true stress-true strain curves of the radial wrought aluminium alloy 6063 during the semi-solid isothermal compression deformation procedure at different isothermal temperatures (i.e., deformation temperatures) are shown in Figure 3. It was revealed that at the same value of true strain, the true stress reduced when the deformation temperature increased. This was because different solid-phase fractions were obtained at different deformation temperatures. At the relatively low deformation temperature, the solid fraction in the sample was high, and the solid grains in the microstructure contacted each other to form a spatial framework structure. This framework structure hindered the slip and rotation of the solid grains. Therefore, the deformation of the sample mainly included the deformation of the spatial framework structure and the squeeze flow of the liquid phase. However, when the deformation temperature increased, the solid phase fraction in the sample decreased with the liquid phase continually surrounding the solid grains, which provided the beneficial condition to the slip and rotation of the solid phase grains. Therefore, the true stress required for the deformation was reduced. Moreover, the true strain required for the stabilization of the true stress was relatively low at a high deformation temperature and a certain strain rate. This was because there were more solid grain agglomerates in the sample when the deformation temperature was relatively low, which required a great amount of deformation to disaggregate these agglomerates. On the contrary, when the deformation temperature was relatively high, fewer agglomerates of the larger solid grains were generated in the sample, which was helpful for the completion of the disaggregation without the disaggregation process of the agglomerates or with only a small amount of deformation to achieve the stabilization of the true stress [30].

It can be concluded from Figure 3a–c that as the true strain increased, the true stress first increased rapidly, then decreased and stabilized, and finally increased slowly. This means that four stages can be observed in the variation of true stress when the true strain increases as shown in Figure 4, which are sharp increase (Stage I), decrease (Stage II), steady state (Stage III), and slow increase (Stage IV).

Stage I: Sharp increase. Semi-solid materials can be regarded as the material with solid grains suspended in the liquid phase, where the solid phase grains or solid-phase grain aggregates contact locally to form a spatial framework structure when the solid phase ratio is large [7,10,30]. At the beginning of the SSIC, the central isostatic stress of the cylindrical compression sample is large, which will force the liquid phase to be squeezed and flowed from the central area to the peripheral area. However, the spatial framework structure, formed by the solid grains, would not be destroyed owing to the small deformation degree, and the deformation of the sample can be recovered if the load is removed. Therefore, the elastic deformation of the spatial framework structure takes place in the sample, namely, the true stress is increased when the true train increased.

Stage II: Decrease. As the SSIC continues, when the space framework structure reaches a certain degree of deformation, its deformation cannot be restored even if the load is removed. In other words, plastic deformation is achieved in the sample. At this stage, the spatial framework structure begins to be destroyed and is gradually and completely surrounded by the liquid phase when the deformation degree increased. The slip and partial rotation of the solid grains then occur at this time. The slip among the solid grains is realized by the shearing force, which promotes the occurrence of holes at the boundary of the grains. These holes would expand rapidly along with the liquid phase, resulting in the damage of samples. This sample damage causes the flow stress to drop rapidly, that is, the true stress drops.

Furthermore, the plastic deformation will also promote the disaggregation of larger solid grain agglomerates, so that the liquid phase wrapped by the agglomerates is released, thereby increasing the liquid content inside the sample. This is helpful for the slippage and rotation of the solid grains, thereby contributing to the reduction of true stress.

Stage III: Steady state. At this stage, the spatial framework structure formed by the solid grains is basically destroyed. The solid particles in the semi-solid material are surrounded by the liquid phase and undergo plastic flow. The deformation resistance is mainly attributed to the slippage and rotation among the solid grains as well as the liquid phase flow. Therefore, the deformation resistance remains basically unchanged, that is, the true stress is at a steady state as the displacement increases.

Stage IV: Slow increase. The main reason for the increase in true stress at this stage is that the true strain is too large, which causes the semi-solid material to roll over during the SSIC process, gradually increasing the contact surface between the pressing head and the sample. This in turn increases the friction between the pressing head and the sample. Furthermore, during the SSIC process of the sample, as the true strain increases, most of the liquid is separated from the solid phases inside the sample and squeezed to the edge of the sample, resulting in an increase in the local solid fraction of the sample. This consequently causes the solid grains to be squeezed and bonded together and therefore generates the plastic deformation [12]. The pressure required to deform the solid grains is greater than the liquid phase flow, thus, it causes the augmentation in the true stress, that is, the true stress gradually enlarges when the true strain is elevated.

The effects of the deformation temperature on the peak stress and steady-state stress are shown in Figure 5. It can be seen that both the peak and steady-state stresses of the aluminum alloy 6063 semi-solid material significantly reduced when the deformation temperature increased at a certain strain rate. This phenomenon indicated that the SSIC of aluminium alloy 6063 was sensitive to the temperature, which is also found by reference [31]. This was because, in the initial stage of SSIC, the liquid fraction in the sample augmented with the increase in the deformation temperature, which increased the thickness of the liquid phase film between the solid grains and reduced the apparent viscosity. This resulted in reduction decrease in the connection strength between the solid grain aggregates. The spatial framework structure can be more easily destroyed, so the peak stress decreased. When it entered the steady state stage, more liquid phase helped the slippage and rotation among the solid grains easier to occur due to the high deformation temperature, leading to a reduction in the steady-state stress. Moreover, the aluminium alloy 6063 semi-solid material was more sensitive to the deformation temperature at the initial stage of SSIC. This was mainly because the peak stress reduced significantly with the elevation in the deformation temperature under different conditions of strain rate. But when the semi-solid 6063 alloys entered the steady-state flow, the steady-state stress tended to stabilize with the enlargement in the deformation temperature when the strain rate increased, which means that the steady-state stress was less sensitive to the deformation temperature under this condition.

The effects of the strain rate on the peak and steady-state stresses of the semi-solid isothermal compression curve of aluminum alloy 6063 are presented in Figure 6. It was found that the peak stress of the semi-solid material increased significantly with the increase in the strain rate at a certain deformation temperature. Nevertheless, a reduction tendency was observed in the steady-state stress with the increase in the strain rate. The explanation for this phenomenon is as follows.(a)At the beginning of the SSIC deformation process, the deformation speed of the sample was small because the strain rate was low, which provided enough time for the liquid phase to participate in repairing the liquid phase film among the deformed solid grains. The deformation of the spatial framework structure of the sample can take place with the slippage and rotation among the solid grains, resulting in a small deformation resistance and small peak stress. On the contrary, the liquid phase did not have enough time to repair the liquid film among the deformed solid grains when the strain rate was high, which caused the difficulty of solid grains to slip and rotate. The solid grains squeezed with each other in this early stage, which hindered the deformation of the sample, thereby, increasing the peak stress.(b)The spatial framework structure of the sample was gradually and completely destroyed with the increase in the semi-solid isothermal compression deformation displacement, which formed solid grains or solid grain agglomerates wrapped by the liquid phase. This influence of the liquid phase on the solid grains or solid grain agglomerates contributed to the occurrence of compression deformation, so the true stress gradually decreased and stabilized. When the strain rate was low, the movement speed of the liquid phase film among the solid grains was small, resulting in the weak shearing effect among the solid grains. Consequently, it was difficult to achieve the disaggregation and destruction of solid grain aggregates. Therefore, higher steady-state stress was required to promote the coordinated deformation between the solid grain agglomerates after entering the steady state flow. On the contrary, when the strain rate was high, the strong shearing effect between the solid grains can promote the disaggregation and destruction of some solid grain aggregates, so that the liquid phase wrapped by the aggregates was released. The liquid content inside the sample, thereby, increased, which was beneficial for the slippage and rotation of the solid grains, so the steady-state stress was observed relatively low.

### 3.3. Constitutive Equations

The constitutive model expresses the material’s dynamic response to the deformation amount, deformation temperature, and the strain rate during the plastic processing, which is crucial for the numerical simulation of the plastic deformation process using the finite element method and can be illustrated by Equation (4) [32],
(4)$$\sigma =f\left(\varepsilon ,\dot{\varepsilon },T\right)$$
where, σ is the flow stress, MPa; ε is the strain of the material; $\dot{\varepsilon }$ is the strain rate, s^−1^.

Semi-solid forming is carried out in the semi-solid temperature range of the metal, where both liquid and solid phases exist. Therefore its deformation behaviour is different from the solid state and the effect of the liquid fraction on the flow stress needs to be considered in the constitutive model [11]. Introducing the liquid phase fraction into Equation (4) to characterize the influence of it, the constitutive model of semi-solid thixotropy of the metal can be expressed as Equation (5),
(5)$$\sigma =f\left(\varepsilon ,\dot{\varepsilon },T,f_{L}\right)$$
where $f_{L}$ is the liquid phase fraction, %.

In the high-temperature hot forming process of the metal, the relationship between its flow stress and the operating parameters including the deformation temperature, strain, and strain rate can be expressed by the Arrhenius equation. The Arrhenius equation can characterize the sensitivity of flow stress to strain rate and deformation temperature as well as having the advantages of concision and a high degree of agreement with actual deformation. Its expressions are shown in Equations (6)–(8) [32,33,34]. In general, the exponential law expressed in Equation (6) is suitable for the stresses low-level values while the exponential law expressed by Equation (7) is applicable for stresses at high-level values. The hyperbolic sine law shown in Equation (8) can be applied to the entire stress range [35,36,37],
(6)$$\dot{\varepsilon }=A_{1}\sigma^{n_{1}}\exp\left(-\frac{Q}{RT}\right)\;\;\;\;\;\;\left(\alpha \sigma <0.8\right)$$
(7)$$\dot{\varepsilon }=A_{2}\exp\left(\beta \sigma \right)\exp\left(-\frac{Q}{RT}\right)\;\;\;\;\;\;\left(\alpha \sigma >1.2\right)$$
(8)$$\dot{\varepsilon }=A{\left[\sin h\left(\alpha \sigma \right)\right]}^{n}\exp\left(-\frac{Q}{RT}\right)\;\;\;\;\;\;\left(\mathrm{all}\;\sigma \right)$$
where *Q* is the deformation activation energy, J/mol; *R* is the molar gas constant; *T* is the deformation temperature, K; *n* and *n*_1_ are the stress exponent; *A*, *A*_1_, and *A*_2_ are factors of the material structure; α and β are stress level parameters, $\alpha =\beta /n_{1}$.

The magnitude of the peak stress is related to the selection of processing equipment and moulds in the semi-solid forming process. In this study, the peak stress is taken as the research object, the liquid phase correction term is introduced into the Arrhenius equation to modify Equation (8). Consequently, the Arrhenius equation adapted to the semi-solid temperature zone can be obtained as Equation (9),
(9)$${\left(1-1.5f_{L}\right)}^{L}\dot{\varepsilon }=A{\left[\sin h\left(\alpha \sigma \right)\right]}^{n}\exp\left(-\frac{Q}{RT}\right)$$
where *L* is the liquid phase factor; ${\left(1-1.5f_{L}\right)}^{L}$ shows that the tendency of the increase in the flow stress due to the enlargement in the strain rate when the liquid phase fraction increases.

In order to obtain the value of α, take the logarithm of Equations (6) and (7) to convert into the linear Equations (10), and (11), respectively:(10)$$\ln\dot{\varepsilon }=n_{1}\ln\sigma +\ln A_{1}-\frac{Q}{RT}\;\;\;\;\;\;\left(\alpha \sigma <0.8\right)$$
(11)$$\ln\dot{\varepsilon }=\beta \sigma +\ln A_{2}-\frac{Q}{RT}\left(\alpha \sigma >1.2\right).$$

Introducing the peak stress and the corresponding strain rate at different deformation temperatures into Equations (10) and (11), the relationship between the strain rate and the peak stress at different deformation temperatures can be obtained, as shown in Figure 7. By linearly fitting the relationship between $\ln\dot{\varepsilon }$ and lnσ as well as that between $\ln\dot{\varepsilon }$ and σ at each deformation temperature shown in Figure 7, the slope of each straight line can be obtained. After calculating the mean value of slopes, it can be obtained that $n_{1}=7.61$ while β=1.32, and consequently $\alpha =\frac{\beta }{n_{1}}=0.1735.$

In order to calculate *n*, Equation (12) can be obtained by taking the logarithm of both sides of Equation (9):(12)$$\ln\dot{\varepsilon }=n\ln\left[\sin h\left(\alpha \sigma \right)\right]+\ln A-\frac{Q}{RT}-L\ln\left(1-1.5f_{L}\right).$$

Importing the peak stress and the corresponding strain rate at different deformation temperatures into Equation (12), the relationship between $\ln\dot{\varepsilon }$ and ln[sinh(ασ)] was obtained and illustrated in Figure 8. The straight lines of the relationship and their slopes can be obtained through the linear fitting method. By calculating the mean value of slopes, it was obtained that n=5.5609.

Based on the results in Section 3.1 as presented in Table 2, the effective liquid fractions at the deformation temperatures of 625 °C, 630 °C, and 635 °C were 11.10%, 13.29%, and 14.42%, respectively. Transforming Equation (12), Equation (13) can be obtained as follows:(13)$$n\ln\left[\sin h\left(\alpha \sigma \right)\right]-\ln\dot{\varepsilon }=-\ln A+\frac{Q}{RT}+L\ln\left(1-1.5f_{L}\right).$$

Therefore, one Equation (13) can be obtained at a certain deformation temperature and a certain strain rate. Correspondingly, 9 equations can be obtained for 3 deformation temperatures and 3 strain rates, where only three parameters *A*, *L*, and *Q* are unknown. The least-squares solution was used to solve the overdetermined equation in this study. These 9 equations can then be written as the following matrix illustrated by Equation (14),
(14)$$DX=\;\left[\begin{array}{ccc} 1 & 1000/T & \ln\left(1-1.5f_{L}\right) \end{array}\right]X=Y$$
where X, Y, D are the 3*1, 9*1, and 9*3 matrixes, respectively. In the matrix X, its elements are $X_{1}=-\ln A$, $X_{2}=\frac{Q}{1000R}$, and $X_{3}=L$. In the matrix Y, $Y_{i}=n\ln\left[\sin h\left(\alpha \sigma \right)\right]-\ln\dot{\varepsilon }$, where i = 1, 2, ⋯ 9.

To calculate the values of parameters *A*, *L*, and *Q*, Equation (14) is multiplied by $D^{T}$ at both sides. Equations (15) and (16) can be obtained by performing the equivalent transformations:(15)$$D^{T}DX=D^{T}Y$$
(16)$$X={\left(D^{T}D\right)}^{-1}D^{T}Y.$$

Introducing the corresponding parameters into Equation (13), matrixes D, X, and Y can be obtained as follows.
$$D=\left[\begin{array}{ccc} 1 & 1.6 & -0.1821\\ 1 & 1.6 & -0.1821\\ 1 & 1.6 & -0.1821\\ 1 & 1.5873 & -0.2223\\ 1 & 1.5873 & -0.2223\\ 1 & 1.5873 & -0.2223\\ 1 & 1.5748 & -0.2437\\ 1 & 1.5748 & -0.2437\\ 1 & 1.5748 & -0.2437 \end{array}\right],X=\left[\begin{array}{c} X_{1}\\ X_{2}\\ X_{3} \end{array}\right],Y=\left[\begin{array}{c} 6.3522\\ 5.4162\\ 5.5404\\ 2.1848\\ 2.4127\\ 3.1344\\ 0.5525\\ -0.2494\\ 0.4351 \end{array}\right]$$

Introducing the above matrixes into Equation (16), the matrix ***X*** can be obtained by performing transposition and inverse matrix operations using Matlab software, which is $X=\left[\begin{array}{c} 60.7210\\ -16.5502\\ 116.0644 \end{array}\right]$. Consequently, the values of parameters were obtained as $A=2.9602\times 10^{70}$, Q=895.641 kJ/mol, L=44.6267. Importing the results of parameters into Equation (9), the peak stress equation of the aluminum alloy in the semi-solid region can be obtained as shown by Equation (17):(17)$${\left(1-1.5f_{L}\right)}^{44.6267}\dot{\varepsilon }=2.9602\times 10^{70}{\left[\sin h\left(0.1735\sigma \right)\right]}^{5.5609}\exp\left(-\frac{895641.8}{RT}\right)$$

It is worth noting that the applicable range of ${\left(1-1.5f_{L}\right)}^{44.6267}$ in Equation (17) is $\left(1-1.5f_{L}\right)>0,$ that is $f_{L}<66.67\%$. This means that Equation (17) is applicable to the semi-solid thixotropic extrusion process when the liquid fraction is less than 66.67%.

The correlation coefficient R and the average absolute relative error (AARE) were used in the study to indicate the accuracy performance of the established peak stress equation of the aluminium alloy in the semi-solid region, which are expressed by Equations (18), and (19), respectively,
(18)$$R_{c}=\frac{\sum_{i=1}^{N}\left(\alpha_{e}-{\bar{\alpha }}_{e}\right)\left(\alpha_{p}-{\bar{\alpha }}_{p}\right)}{\sqrt{\sum_{i=1}^{N}{\left(\alpha_{e}-{\bar{\alpha }}_{e}\right)}^{2}\cdot \sum_{i=1}^{N}{\left(\alpha_{p}-{\bar{\alpha }}_{p}\right)}^{2}}}$$
(19)$$\mathrm{AARE}\left(\%\right)=\frac{1}{N}\sum_{i=1}^{N}\left|\frac{\alpha_{e}-\alpha_{p}}{\alpha_{e}}\right|\times 100\%$$
where $\alpha_{e}$ is the test value of the peak stress, MPa; ${\bar{\alpha }}_{e}$ is the average value of the tested peak stress, MPa; $\alpha_{p}$ is the predicted value of peak stress, MPa; ${\bar{\alpha }}_{p}$ is the average value of the predicted peak stress, MPa; *N* is the statistical number of data points.

As shown in Figure 9, the correlation coefficient between the predicted value of the peak stress and the experimentally tested value was found as 0.9796. The average relative error was obtained as 5.01% based on Equation (19), reflecting a high accuracy. This means that the constitutive equation can accurately express the change in the peak stress of aluminium alloy 6063 in the semi-solid region, which can be used for selecting the forming equipment of the semi-solid metal forming process.

### 3.4. Deformation Mechanism

#### 3.4.1. Macroscopic Results and Analysis

The macroscopic photo of the radially forged aluminium alloy 6063 semi-solid isothermal compression samples is shown in Figure 10. Figure 10a presents the macroscopic photo of the samples processed at the strain rate of 0.1 s^−1^ with different deformation temperatures. It was found that the compressed sample gradually broke with the elevation in the deformation temperature. The higher the deformation temperature, the more obvious the degree of crushing of the compressed sample. This was because, at the beginning of the semi-solid isothermal compression deformation, the liquid phase already existed in the aluminium alloy 6063 semi-solid material, the fraction of which elevated with the enlargement in the deformation temperature. The liquid phase in the compressed sample was squeezed and flew to the sidewall of the sample at the central region and caused the sidewall to rupture during the SSIC deformation process. This liquid phase in the compressed sample continued to flow along the rupture of the sidewall in the final stage of the SSIC deformation, resulting in the fragmentation of the entire compressed sample. Nevertheless, the phenomenon aforementioned did not occur in the actual SSMF process. This was mainly because the mould cavity in the actual SSMF process was closed, which consequently avoided the outflow of the liquid phase [38,39].

The macroscopic photo of the samples processed at the deformation temperature of 630 °C under different strain rates is shown in Figure 10b. It was found that the macroscopic photo of the compressed sample changed significantly when the strain rate increased. The integrity of the compressed sample was relatively high at a relatively small strain rate, while the compressed sample was broken as the strain rate increased. This was because the flow of the liquid phase inside the semi-solid material became more intense with the elevation in the strain rate, so the degree of crushing increased.

According to the principle of pressure processing [13,32], the amount of deformation inside the sample is uneven in the process of material compression and deformation. The deformation area can be divided into hard deformation region, huge deformation region, and free deformation region, as shown in Figure 11a. The macroscopic photo of the section view along the longitudinal direction of the sample processed by the SSIC deformation is presented in Figure 11b, where the strain rate was 0.1 s^−1^, the deformation temperature was set as 630 °C, and the strain was 0.7. It was found that similar to the compression deformation of solid metal, the semi-solid isothermal compression deformation sample also exhibited three different deformation regions, which are the hard deformation area near the end (Region A), the huge deformation area in the centre (Region B), and the free deformation region (Region C) in the convex region.

#### 3.4.2. Microstructure and Deformation Mechanism Analysis

The microstructures from different positions of the semi-solid compressed sample are shown in Figure 12. Comparing Figure 12b–f, the liquid fraction in the structure tends to increase from the hard deformation region (Region A) to the huge deformation region (Region B), then to the free deformation region (Region C). A large increase in the liquid fraction was found in the free deformation region (Region C), compared to the huge deformation region (Region B). The most serious liquid segregates were observed in the free deformation region (Region C). This was because the liquid phase in the microstructure at Position 1 as shown in Figure 12a was squeezed out from the space among solid grains and in turn, flew to Positions 2 and 3 forced by the compression of the deformation as the amount of compression deformation continued to increase. Furthermore, under the action of the deformation force, the liquid phase at the centre of the sample (Positions 2 and 3) gradually flew to the edge position (Position 4 or 5), resulting in the uneven distribution of solid and liquid phases in each deformation region in the compressed sample.

The liquid phase of the semi-solid material was mainly distributed at the boundary of the solid grains, which has the effect of isolating solid grains. When the compression force was applied to the semi-solid material, the separated flow of liquid and solid phases would take place. Generally, the liquid content has a tendency to move before the solid grains. The reduction of the liquid phase would cause the subsequent flow of solid grains to contact with each other. If the deformation continues, severe plastic deformation would take place in the solid grains. Furthermore, the flow of the liquid phase would also cause the liquid phase segregation. The deformation was restricted in the deformation region. Therefore, the microstructure basically remained the original semi-solid microstructure without obvious segregation, as shown in Figure 12b. However, the liquid fraction was observed slightly reduced and the bonding phenomenon was found among part of the solid grains. This means that in the process of compression deformation, the mixed flow of solid grains and the liquid phase was the main phenomenon (i.e., the grain sliding or the grain rotation along with the liquid phase) in the hard deformation region. Partial liquid phase flow also occurred in Region A under the squeeze action of the deformation force. The deformation amount in Region B was observed relatively large, as shown in Figure 12c,d. In Region B, the solid grains appeared to be squashed with liquid phase segregation observed in some regions. Therefore, the liquid flow is the major phenomenon in the huge deformation region (i.e., Region B) at the beginning of SSIC. This liquid content was squeezed out with the increase in the compression deformation. The solid grains contacted each other and were squeezed by the deformation force, resulting in the plastic deformation of the solid grains. Meanwhile, a certain degree of slip and rotation appeared among the solid grains.

The nearly spherical phenomenon was observed in solid grains in the free deformation region (see Positions 4 and 5), but a more serious liquid phase segregation was also found as shown in Figure 12e,f. This was mainly because the liquid phase from the hard and huge deformation regions flew into the free deformation region, resulting in the macroscopic transfer of the liquid. Therefore, the closer to the free-deformation region, the more liquid phase there was, the more serious the liquid phase segregation. Moreover, the thorough partial solid-liquid separation was also observed in the free deformation region because of the movement of the liquid phase, where pores would be generated if the surrounding liquid or solid phase cannot be replenished in time. Therefore, liquid flow and the sliding or rotation of solid grains along the liquid phase are the deformation modes in the free deformation region.

Based on the analysis mentioned above, the deformation mechanism of the radially forged aluminium alloy 6063 material during the SSIC mainly included the grain slide or grain rotation along with the liquid film, the sliding among solid grains, and the plastic deformation of solid grains. Moreover, according to the analysis of the microstructure in different deformation regions, during the SSIC, different degrees of liquid phase flow occurred in all these three deformation regions. Considering this liquid phase flow, there were four deformation mechanisms in SSIC, which is presented in Figure 13, where the large blue arrow shows the compression direction during the SSIC deformation process, the little red arrow indicates the direction of grain movement, while the square and the rectangle indicates the deformation area before, and after, the deformation, respectively.

As presented in Figure 13a, the highest liquid phase fraction can be obtained in the liquid phase flow mechanism. The thick liquid film wraps the solid-phase grains and no contact existed among the solid grains. Consequently, the flow channel for the liquid phase is unobstructed. The liquid phase flows perpendicular to the compression direction while the solid grains mostly flow parallel to the compression direction when compression deformation is performed. The liquid phase segregation would then take place after the deformation. As illustrated in Figure 13b, the liquid fraction reduces and the nearly spherical solid particles are enfolded by a relatively thin liquid film in the mechanism of grain slide or grain rotation along with the liquid film. When the compression deformation is carried out, solid grains flow both perpendicular and parallel to the direction of the compression force. This means that the solid grains slide or rotate along with the liquid film, forming a mixed flow of solid grains and liquid phase.

It can be found in Figure 13c that, in the sliding mechanism, the liquid phase fraction further decreased while the thickness of the liquid film among the solid grains continued to reduce. Solid grain clusters were formed as some solid grains contact each other, which blocks the flow channel for the liquid phase. Under the compression deformation, the solid grains inside the solid grain cluster would slide relative to the adjacent solid grains due to the shear force [7,12,40,41]. As presented in Figure 13d, a low phase fraction of the liquid phase is observed in the mechanism of the plastic deformation of solid grains while little liquid film among the solid grains can also be observed. In this mechanism, most of the solid grains contact each other, forming the solid grain clusters. The sliding among the solid grains is not intense enough to generate a large plastic deformation during the compression deformation process. Therefore, the plastic deformation would take place on the solid grains to achieve a large SSIC deformation of the metal material.

In summary, the phase fraction of liquid determines the deformation mode of the semi-solid material during the isothermal compression deformation process. When the phase fraction of liquid is large, the content of the liquid phase in the microstructure is high and thus, solid grains would not completely contact each other. The flow channel of the liquid phase is therefore unobstructed. The deformation mode mainly contains the liquid phase flow and the sliding or rotation of the solid grain along with the liquid film. When the phase fraction of liquid is small, the content of the liquid phase in the microstructure reduces, resulting in the local contact of solid grains. The flow channel of the liquid phase is then hindered. The deformation mode then mainly includes the sliding among solid grains and the plastic deformation of solid grains. Furthermore, the order of the main deformation mechanism with the decrease in the liquid phase fraction is from the liquid phase flow to the grain slide or grain rotation along with the liquid film, then to the sliding among the solid grains, and finally to the plastic deformation of solid particles. In terms of a specific compressed sample, the grain slide or grain rotation along the liquid phase is the main deformation mode in the hard deformation region (Region A); the plastic deformation of solid grains is the main deformation mode in the huge deformation region (Region B), while a certain degree of sliding among the solid grains also appears in this region; the liquid phase flow and the grain slide or grain rotation along the liquid film are the main deformation modes in the free deformation region (Region C).

## 4. Conclusions

In this work, SSIC is proposed to investigate the microstructure, deformation characteristics, and deformation mechanism of semi-solid alloy, where the cold radial forging is introduced in the SIMA process to prepare the high-quality semi-solid billet. The main conclusions drawn from this study are shown as follows.

(1) The average grain size, shape factor, and effective liquid phase fraction increase with the enhancement in the deformation temperature. The high-quality semi-solid alloy, which has the average grain size in the range of 59.22–73.02 μm while the average shape factor in the range of 0.71–0.78, can be obtained when the cold radial forged 6063 aluminum alloy is treated by SSIT process at the deformation temperature in the range of 625–635 °C.

(2) The true stress-true strain curve of the cold radial forged 6063 aluminium alloy SSIC mainly includes four deformation stages: sharp increase stage, decrease stage, steady state stage, and slow increase stage. The true stress decreases when the deformation temperature increases at a certain strain rate, which means that both the peak and steady-state stresses decrease with the enlargement in the deformation temperature. The peak stress elevates while the steady-state stress decreases with the enlargement in the strain rate at a certain deformation temperature.

(3) Introducing the liquid phase correction term $S={\left(1-1.5f_{L}\right)}^{L}$ into the Arrhenius equation, the peak stress constitutive equation of the semi-solid region of aluminium alloy 6063 in the semi-solid deformation temperature range is calculated and established as ${\left(1-1.5f_{L}\right)}^{44.6267}\dot{\varepsilon }=2.9602\times 10^{70}{\left[\sin h\left(0.1735\sigma \right)\right]}^{5.5609}\exp\left(-\frac{895641.8}{RT}\right)$. The correlation coefficient between the predicted value of the peak stress by the constitutive equation and the experimental result is 0.9796 while the average relative error is 5.01%, which means that the constitutive equation established in this study can accurately express the change in the peak stress of aluminium alloy 6063 in the semi-solid temperature region with the variation of the effective liquid phase fraction, deformation temperature, strain rate, and the strain.

(4) The deformation mechanisms of cold radial forged aluminium alloy 6063 during SSIC mainly include four modes: liquid phase flow, grain slide or grain rotation along with the liquid film, sliding among solid grains, and the plastic deformation of solid grains. For a specific sample, the grain slide or grain rotation along the liquid film is the main deformation mode in the hard deformation region. In the huge deformation region, the plastic deformation of solid grains is the main deformation mode while a certain degree of sliding among solid grains also exists. The liquid phase flow and the grain slide or grain rotation along the liquid film are the main deformation mode in the free deformation region.

## Figures and Tables

**Figure 1 materials-14-00194-f001:**
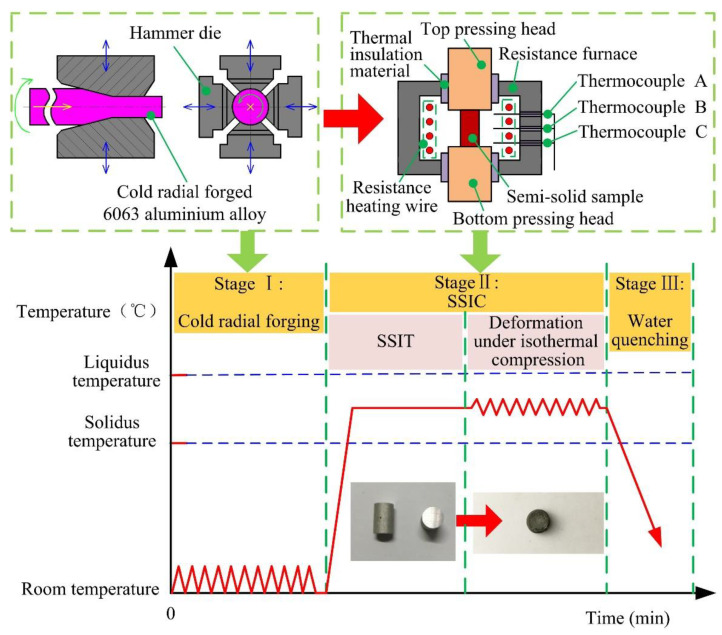
Schematic illustration of the SSIC process of the cold radial forged 6063 aluminium alloy.

**Figure 2 materials-14-00194-f002:**
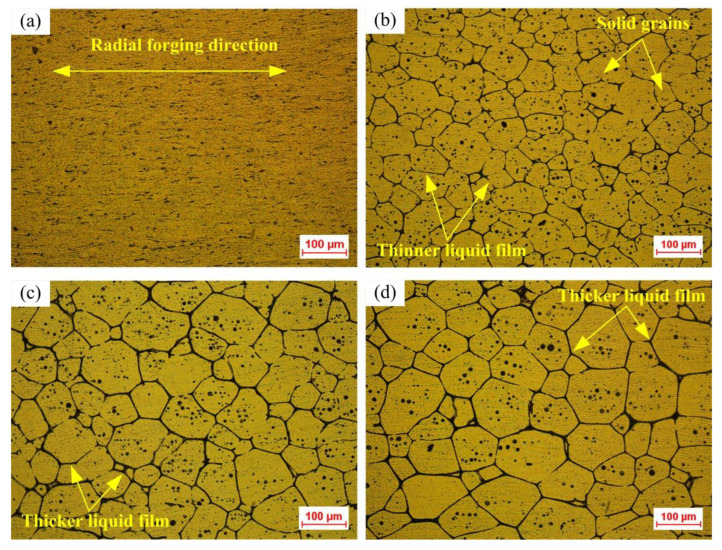
Microstructure of; (**a**) the cold radial forged 6063 aluminium alloy without SSIT, and semi-solid 6063 aluminium alloy after SSIT with the isothermal temperatures of; (**b**) 625 °C, (**c**) 630 °C, and (**d**) 635 °C.

**Figure 3 materials-14-00194-f003:**
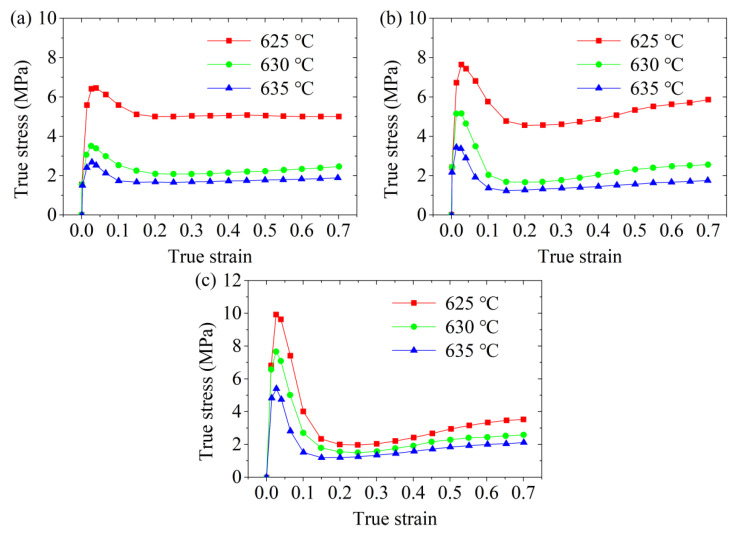
Ture stress-true strain curves of the cold radial forged 6063 aluminium alloy processed by the semi-solid isothermal compression at different temperatures with strain rates of (**a**) 0.01 s^−1^, (**b**) 0.1 s^−1^, and (**c**) 1 s^−1^.

**Figure 4 materials-14-00194-f004:**
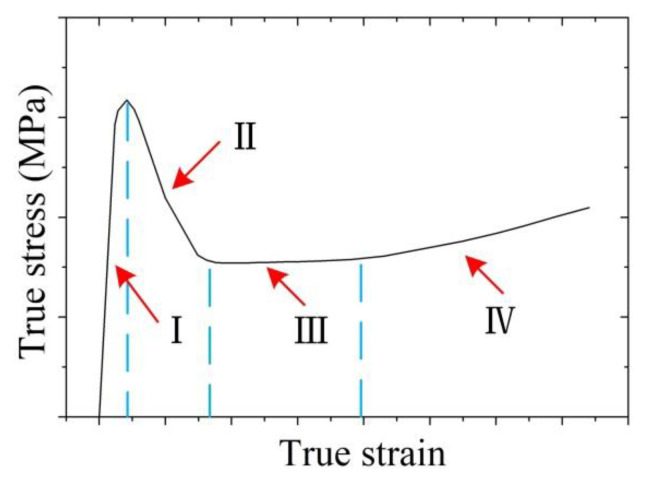
The deformation pattern of the cold radial forged 6063 aluminium alloy processed by SSIC.

**Figure 5 materials-14-00194-f005:**
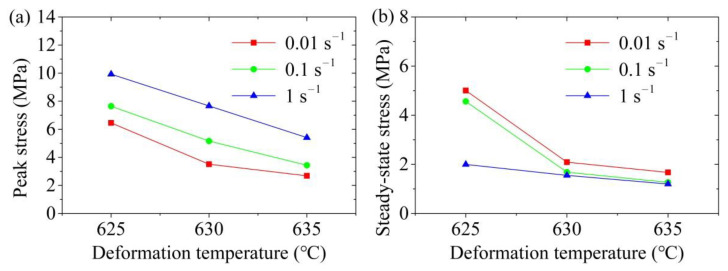
Influence of the deformation temperature in SSIC on the (**a**) peak stress, and (**b**) steady-state stress.

**Figure 6 materials-14-00194-f006:**
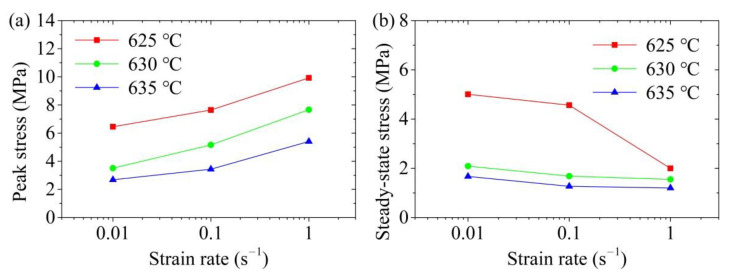
Influence of the strain rate in SSIC on the (**a**) peak stress; and (**b**) steady-state stress.

**Figure 7 materials-14-00194-f007:**
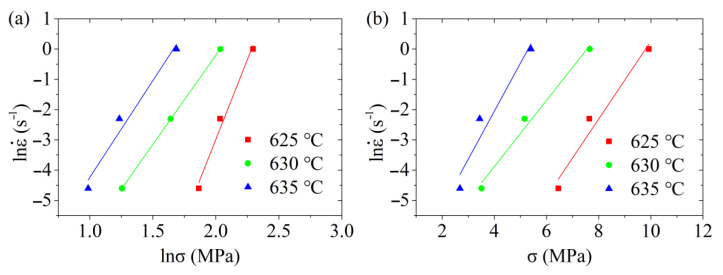
Relationship between the natural logarithm of the strain rate, and (**a**) the natural logarithm of the peak stress, and (**b**) the peak stress.

**Figure 8 materials-14-00194-f008:**
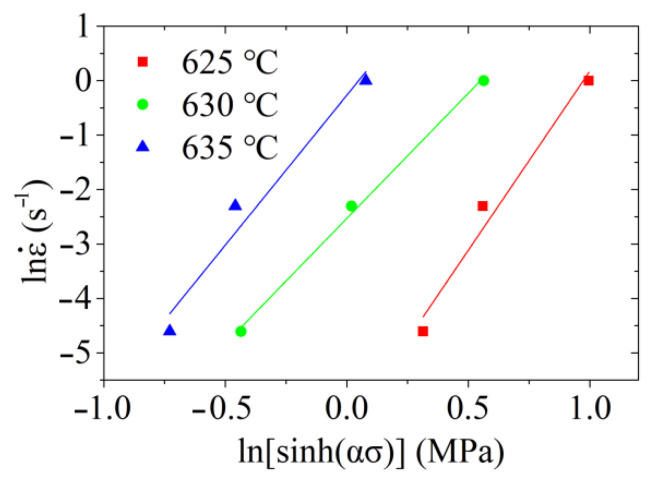
Relationship between $\ln\dot{\varepsilon }$ and ln[sinh(ασ)].

**Figure 9 materials-14-00194-f009:**
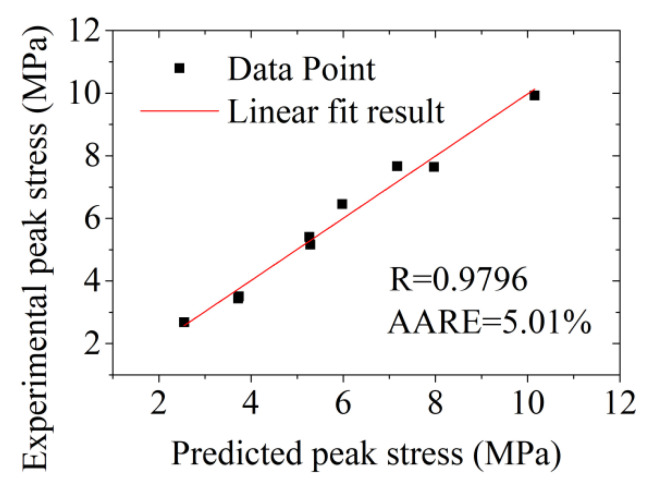
Relationship between the experimental and calculated peak stress of the cold radial forged 6063 aluminum alloy processed by SSIC.

**Figure 10 materials-14-00194-f010:**
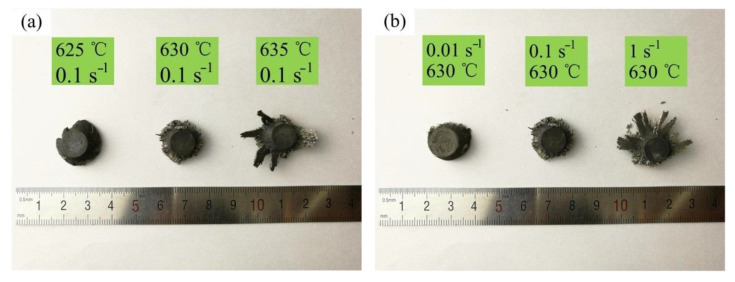
Macroscopic photo of samples after the SSIC at different, (**a**) deformation temperatures, and (**b**) strain rates.

**Figure 11 materials-14-00194-f011:**
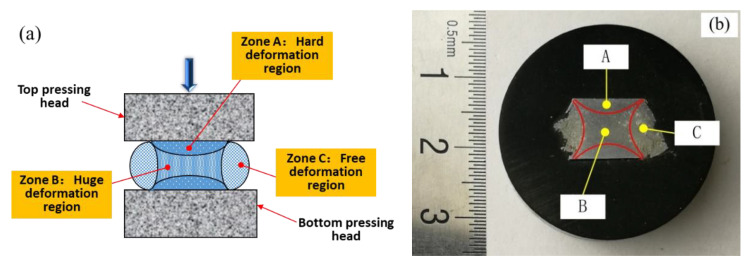
Deformation characteristics of the compressed samples: (**a**) Deformation region division, and (**b**) the longitudinal macroscopic photo of samples after the semi-solid isothermal compression (deformation temperature: 630 °C, strain rate: 0.1 s^−1^).

**Figure 12 materials-14-00194-f012:**
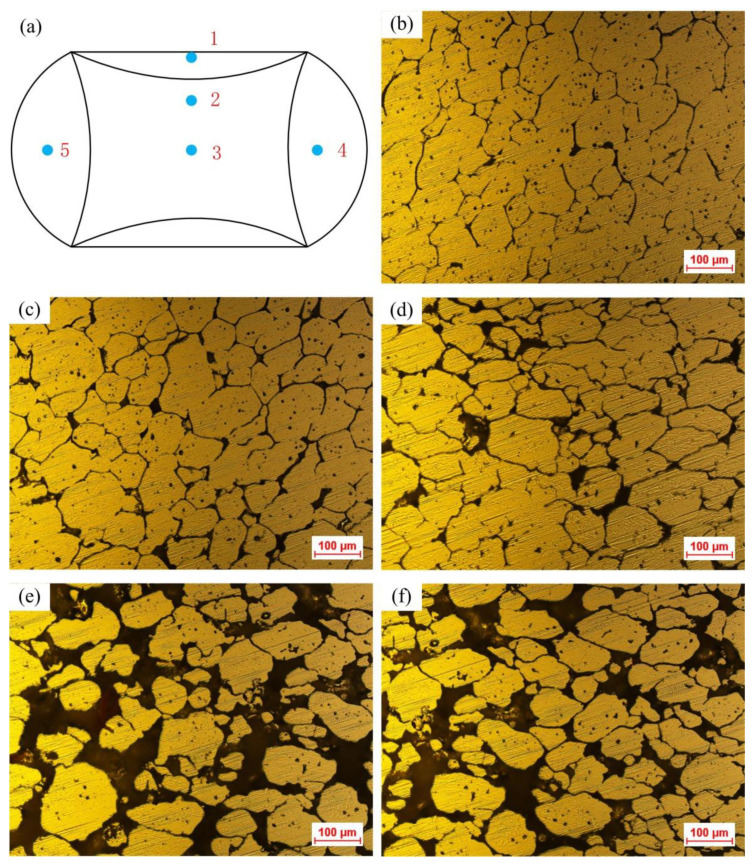
Microstructures of the cold radial forged 6063 alloy processed by SSIC (deformation temperature: 630 °C, strain rate: 0.1 s^−1^): (**a**) Sample positions, (**b**) Position 1 in Region A, (**c**) Position 2 in Region B, (**d**) Position 3 in Region B, (**e**)Position 4 in Region C, and (**f**) Position 5 in Region C.

**Figure 13 materials-14-00194-f013:**
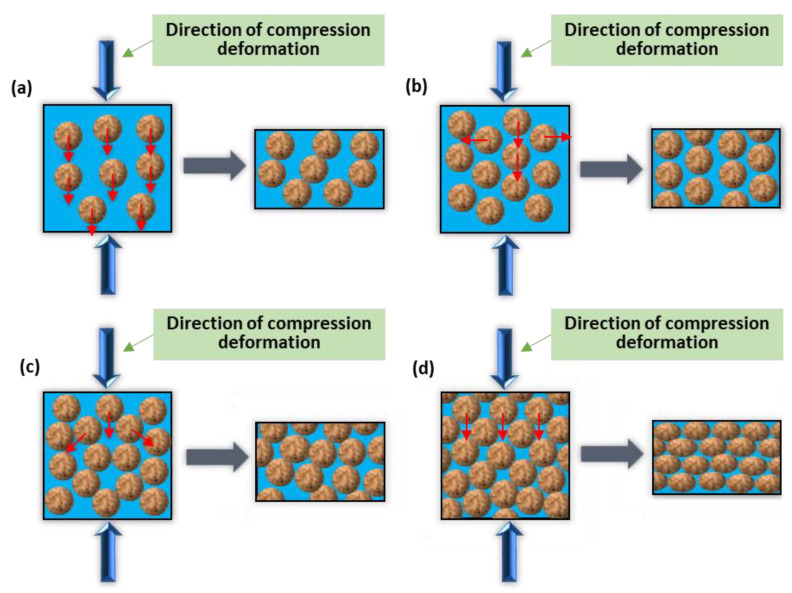
Schematic diagram of deformation mechanisms: (**a**) Liquid phase flow, (**b**) solid grains slip or rotate along the liquid film, (**c**) slipping among solid grains, and (**d**) plastic deformation of solid grains.

**Table 1 materials-14-00194-t001:** Chemical composition of the 6063 aluminium alloy (wt.%).

Mg	Si	Mn	Zn	Fe	Cu	Ti	Cr	Al
0.51	0.39	0.03	0.02	0.15	0.01	<0.001	<0.005	Bal.

**Table 2 materials-14-00194-t002:** Experimental process parameters.

Experimental Number	Isothermal Temperature (°C)	Strain Rate (s^−1^)	Strain	Isothermal Holding Time (min)
1	625	0	0	10
2	630	0	0	10
3	635	0	0	10
4	625	0.01	0.7	10
5	625	0.1	0.7	10
6	625	1	0.7	10
7	630	0.01	0.7	10
8	630	0.1	0.7	10
9	630	1	0.7	10
10	635	0.01	0.7	10
11	635	0.1	0.7	10
12	635	1	0.7	10

**Table 3 materials-14-00194-t003:** Microstructural characteristics of 6063 alloy microstructure processed by SSIT at different isothermal temperatures.

Isothermal Temperature (°C)	Average Grain Size (μm)	Average Shape Factor	Effective Iiquid Fraction
625	59.22	0.71	11.10%
630	64.11	0.75	13.29%
635	73.02	0.78	14.42%

## Data Availability

The data presented in this study are available on request from the corresponding author. The data are not publicly available due to project confidentiality requirements.

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
