# Peer review of "Deformation Characteristics and Constitutive Equations for the Semi-Solid Isothermal Compression of Cold Radial Forged 6063 Aluminium Alloy"

_materials, 2021, doi:10.3390/ma14010194_

Round 1

Reviewer 1 Report

The paper is presented with reasonable experimental data and explained them in detail. But in some places the modifications can be done by adding or correcting the contents.

1. In line nos. 24 and 25, try to put their ranges like in scientific form. For ex. in line 60. Use the similar types for mentioning its working range.

2. As 60% of the paper is about flow stress modeling, it is good to add more relevant references about the constitutive modeling. Here in this paper, Arrhenius-type constitutive model is used. So add some papers related to this equation.

3. In line 97, put temperature units for both values (615 degree Celsius and 655 Celsius).

4. Equation (3) is missing in the paper. check whether you missed the numbering.

5. In line nos. 149 and 346, put temperature units for 625, 630.

6. It would be better if you add the experimental vs prediction curves for flow stress values. Figure.9 is enough make a correlation in terms of numerical and graphical validations. But, somehow, R-square value is based on the number of samples and the range of experimental values. It can be altered by changing the samples and the experimental values. So please use Figure 3 (experimental values) and combine the prediction values with that plot, then make a comparison plot for the better illustration of your proposed model.

overall, the paper tends to have some grammatical errors such as article missing, commas, etc. So try to check one more time and correct it before re-submission. Thank You.

Author Response

The authors would like to take this opportunity to sincerely thank the reviewer for his/her valuable comments. We have further revised the manuscript according to the reviewer's comments. We hope the revised manuscript will meet with your approval. The detailed responses to the reviewer’s comments are provided below.

Comments and Suggestions for Authors

The paper is presented with reasonable experimental data and explained them in detail. But in some places the modifications can be done by adding or correcting the contents.

  1. In line nos. 24 and 25, try to put their ranges like in scientific form. For ex. in line 60. Use the similar types for mentioning its working range.

Response: Thanks for pointing this out. We have revised the ranges wrote in scientific form. The revised content is shown as follows.

Results show that the average grain size in the range from 59.22 to 73.02 μm while the average shape factor in the range from 0.71 to 0.78 can be obtained in the microstructure after the cold radial forged 6063 alloys were treated with SSIT process.

Please see lines 24-27 on page 1 in the revised manuscript.

  1. As 60% of the paper is about flow stress modeling, it is good to add more relevant references about the constitutive modeling. Here in this paper, Arrhenius-type constitutive model is used. So add some papers related to this equation.

Response: Thanks for the reviewer’s comment. More relevant references about the constitutive modeling have been added in the revised manuscript. Papers related to the Arrhenius-type constitutive model have also been added in the paper. The references added are shown as follows.

  1. Meshkabadi, R.; Pouyafar, V.; Javdani, A.; Faraji, G. An Enhanced Steady-State Constitutive Model for Semi-solid Forming of Al7075 Based on Cross Model. Metall. Mater. Trans. A. 2017, 1-11.
  2. Wang, J.J.; Phillion, A.B.; Lu, G.M. Development of a visco-plastic constitutive modeling for thixoforming of AA6061 in semi-solid state. J. Alloys. Compd. 2014, 609, 290-295.
  3. Lu, Y.l.; Li, M.Q.; Li, X.C. Deformation Behavior and Constitutive Equation Coupled the Grain Size of Semi-Solid Aluminum Alloy. J. Mater. Eng. Perform. 2010, 19, 1337-1343.
  4. Li, Y.; Zhao, S.; Fan, S.; Yan, G. Study on the material characteristic and process parameters of the open-die warm extrusion process of spline shaft with 42CrMo steel. J. Alloys. Compd. 2013, 571, 12-20, doi:10.1016/j.jallcom.2013.03.209.
  5. Changizian, P.; Zarei-Hanzaki, A.; Roostaei, A.A. The high temperature flow behavior modeling of AZ81 magnesium alloy considering strain effects. Mater. Design 2012, 39, 384-389.
  6. Ashtiani, H.R.R.; Parsa, M.H.; Bisadi, H. Constitutive equations for elevated temperature flow behavior of commercial purity aluminum. Materials ence & Engineering A 2012, 545, 61-67.

Please see References 14-16 on page 21 and References 35-37 on page 22 in the revised manuscript.

  1. In line 97, put temperature units for both values (615 degree Celsius and 655 Celsius).

Response: Thanks for the comment. It has been revised as suggested. The revised content is shown as follows.

The results show that the solidus and liquidus temperatures of this alloy are 615°C and 655°C, respectively.

Please see lines 102 on page 3 in the revised manuscript.

  1. Equation (3) is missing in the paper. check whether you missed the numbering.

Response: Thanks for pointing out this. The number has been added for Equation (3) in the revised manuscript.  All equations haven been checked in the revised manuscript. The revised content is shown as follows.

                                 (3)

Please see line 151 on page 5 in the revised manuscript.

  1. In line nos. 149 and 346, put temperature units for 625, 630.

Response: Thanks for pointing this out. Temperature unit has been added as suggested. The revised content is shown as follows.

Figures 2(b) - (d) show the semi-solid microstructures obtained after the radial forged alloy processed by SSIT at the isothermal temperature of 625˚C, 630˚C, and 635 ˚C for 10 min, respectively.

Based on the results in Section 3.1 as presented in Table 2, the effective liquid fractions at the deformation temperatures of 625˚C, 630˚C, and 635˚C were 11.10%, 13.29%, and 14.42%, respectively.

Please see line 160 on page 6 and line 381 on page 13 in the revised manuscript.

  1. It would be better if you add the experimental vs prediction curves for flow stress values. Figure.9 is enough make a correlation in terms of numerical and graphical validations. But, somehow, R-square value is based on the number of samples and the range of experimental values. It can be altered by changing the samples and the experimental values. So please use Figure 3 (experimental values) and combine the prediction values with that plot, then make a comparison plot for the better illustration of your proposed model.

Response: Thanks for the reviewer’s comment. The constitutive equation established in this paper is used for the calculation of the peak stress instead of the true stress as the peak stress is an important factor for selecting the forming equipment of the semi-solid metal forming process. So only the peak stresses in Figure 3 are used in Figure 9 for the linear fitting. Since the constitutive equation is used for calculating the peak stress, other stress data in Figure 3 is not appropriate for the comparison. However, it is worthy to establish the constitutive equation for calculating the flow stress in the whole investigated range, which will be carried out in our future research.

  1. Overall, the paper tends to have some grammatical errors such as article missing, commas, etc. So try to check one more time and correct it before re-submission. Thank You.

Response: Thanks for the reviewer’s comment. The grammatical errors have been double-checked in the revised manuscript.

Reviewer 2 Report

Dear Authors, 

Congratulations on your research, yet I find that the terminology used in the paper is not appropriate and does not do justice to your work.
Please perform a thorough English language revision and check the terms used.
In the attached document you find several recommendations. 

My best regards.

Author Response

The authors would like to take this opportunity to sincerely thank the reviewer for his/her valuable comments. We have further revised the manuscript according to the reviewer's comments. We hope the revised manuscript will meet with your approval. The detailed responses to the reviewer’s comments are provided below.

Comments and Suggestions for Authors

Dear Authors, 

Congratulations on your research, yet I find that the terminology used in the paper is not appropriate and does not do justice to your work.
Please perform a thorough English language revision and check the terms used.
In the attached document you find several recommendations. 

My best regards.

Response: Thanks for the reviewer’s comments. We have revised the manuscript according to the attached document. The responses to comments from the reviewer’s document are shown as follows.

  1. alloys used.

Response: Thanks for the reviewer’s comment. “materials employed” has been revised as “alloys used”. The revised contents are shown as follows.

Al-Mg-Si based alloys are popular alloys used in the automotive industry.

Please see line 16 on page 1 in the revised manuscript.

  1. mechanism of what?

Response: Thanks for pointing out this. It has been revised as “deformation mechanism”. The revised contents are shown as follows.

However, limited studies have been performed to investigate the microstructure, deformation characteristics, and deformation mechanism for the semi-solid 6063 alloys.

Please see line 18 on page 1 in the revised manuscript.

Moreover, little research has been performed on the microstructure, deformation characteristics, and deformation mechanism for the semi-solid 6063 alloys.

Please see line 88 on page 2 in the revised manuscript.

In this paper, there was a total of 12 semi-solid isothermal compression experiments performed to examine the effects of the operating parameters on the microstructure, deformation characteristics, and the deformation mechanism of the cold radial forged 6063 aluminum alloy.

Please see line 130 on page 5 in the revised manuscript.

  1. an

Response: The word “the” has been revised as “an”. The revised contents are shown as follows.

Results show that an average grain size in the range from 59.22 to 73.02 μm and average shape factor in the range from 071 to 078 can be obtained in the microstructure after the cold radial forged 6063 alloys were treated with SSIT process.

Please see line 24 on page 1 in the revised manuscript.

  1. and

Response: Thanks for pointing out this. “while the” has been revised as “and an”. The revised contents are shown as follows.

Results show that an average grain size in the range from 59.22 to 73.02 μm and an average shape factor in the range from 071 to 078 can be obtained in the microstructure after the cold radial forged 6063 alloys were treated with SSIT process.

Please see line 25 on page 1 in the revised manuscript.

  1. “has fine and spherical microstructures” is unclear, rephrase. What does dense interior mean?

Response: Thanks for pointing out this. We have rephrased this sentence. The revised contents are shown as follows.

In the SSMF process, the semi-solid billet, which has small and near-spherical microstructures, is used to fabricate high-quality components with a dense internal structure, accurate size, few macro defects, and few micro defects.

Please see lines 48-50 on page 2 in the revised manuscript.

  1. Table 1 does not show that. Rephrase.

Response: Thanks for pointing out this. The content has been rephrased as suggested. The revised contents are shown as follows.

The solidus and liquidus temperatures of this alloy were identified by the heat flow-temperature curve, which was obtained by using an ‘STA 449F5’ differential scanning calorimeter (DSC). Samples with the dimension of 6mm×6mm×5mm were heated from room temperature to 750°C at 10°C/min under nitrogen atmosphere, and analyzed by DSC. The results show that the solidus and liquidus temperatures of this alloy are 615°C and 655°C, respectively.

Please see line 98-102 on page 3 in the revised manuscript.

  1. Using A) B) C) and Stage I, II, III is redundant. Use only one notation.

Response: Thanks for the reviewer’s comment.  “A) B) C)” has been deleted in the revised manuscript. The revised contents are shown as follows.

Stage I: The barstock of 6063 alloys with a diameter of 100 mm was deformed by the cold radial forging process until the reduction in the area reached 64% (i.e.φ60mm). Then the sample was finished from the cold radial forged alloy along the direction of radial forging, which had the diameter and height of 10 mm and 15 mm, respectively. Stage II: The SSIC process including SSIT and the deformation of isothermal compression were carried out with the SSIC- equipment which mainly included the top pressing head, bottom pressing head, insulation material, resistance furnace, and thermocouples. During the SSIC process, the resistance furnace was heated up to the setting deformation temperature, and then the cold radial forged sample was placed between the top and bottom pressing head for SSIT. The semi-solid samples can be obtained when the SSIT process had been executed for 10 min at different isothermal temperatures. After that, these samples were compressed with the true strain of 0.7 to obtain the compressed semi-solid sample by the top and bottom pressing head at different deformation temperatures and different strain rates. For all experiments, the deformation temperature was selected the same as the isothermal temperature in SSIT. Stage III: The compressed semi-solid sample was cooled rapidly by water quenching for preserving the microstructures of the sample.

Please see lines 106-124 on page 3 in the revised manuscript.

  1.  

Response: Thanks for pointing out this. “sampling” has been revised as “the action of sampling”. The revised contents are shown as follows.

Then the action of sampling was finished from the cold radial forged alloy along the direction of radial forging

Please see line 110 on page 3 in the revised manuscript.

  1. Abestos is ued as an insulating material, and perhaps it would be better if you did not state that it is asbestos since it is banned in most countries

Response: Thanks for pointing out this. “Abestos” has been changed as “insulation material”. The revised contents are shown as follows.

The SSIC process including SSIT and the deformation of isothermal compression were carried out with the SSIC- equipment which mainly included the top pressing head, bottom pressing head, insulation material, resistance furnace, and thermocouples.

Please see line 115 on page 3 and Figure 1 on page 4 in the revised manuscript.

  1.  

Response: Thanks for pointing out this. “designed” has been changed as “selected”. The revised contents are shown as follows.

The deformation temperature was selected the same as the isothermal temperature in SSIT.

Please see line 122 on page 3 in the revised manuscript.

In Experiments 4-6, the isothermal temperatures were all selected as 625 ˚C at the strain rate of 0.01, 0.1, and 1 s-1, respectively

Please see line 135 on page 5 in the revised manuscript.

  1. By water quenching.

Response: Thanks for pointing out this. “with the water quenching process” has been changed as “by water quenching”. The revised contents are shown as follows.

The compressed semi-solid sample was cooled rapidly by water quenching for preserving the microstructures of the sample.

Please see line 123 on page 3 in the revised manuscript.

  1.  

Response: Thanks for pointing out this. The content in Figure 1 has been revised. The details are shown as follows.

Please see Figure 1 on page 4 in the revised manuscript.

  1. Stress and strain are not measured directly. perhaps: were determined in-situ by the SSIC-equipment.

Response: Thanks for the reviewer’s comment. The content has been revised in the paper. The revised sentence is shown as follows.

During the semi-solid isothermal compression experiments, the true stress and true strain are determined in-situ by the SSIC-equipment.

Please see lines 139-141 on page 5 in the revised manuscript.

  1. Some contents are suggested to be deleted, such as “When the experiment had been finished”, “which was”, “the”, “was”, “After the polish”, “s”, “using the”.

Response: The content has been revised as suggested. The revised contents are shown as follows.

Samples were taken along the axial direction, which was first ground through the sandpapers with the grits of 200, 400, 600, 800, 1000, 1200, and 1500 in sequence and then was polished through the diamond paste with the particle size of 0.1 μm. Samples were taken along the axial direction, first ground through the sandpapers with the grits of 200, 400, 600, 800, 1000, 1200, and 1500 in sequence and then polished through the diamond paste with the particle size of 0.1 μm. Samples were etched in the aqueous HF solution with a concentration of 5% for about 90 s at room temperature.

Please see lines 141-146 on page 5 in the revised manuscript.

  1.  

Response: “grain” has been revised as “grains”. The revised content is shown as follows.

where A, N, P are the area, number, perimeter of solid grains, respectively;

Please see line 151 on page 5 in the revised manuscript.

  1.  

Response: Thanks for pointing out this. “directionality” has been revised as “texture”. The revised contents are shown as follows.

It can be seen from Figure 2(a) that an obvious texture can be observed in the microstructure.

Please see line 156 on page 6 in the revised manuscript.

  1. “The phase structure and impurities were all distributed along the radial forging direction.” was deleted.

Response: Thanks for the reviewer’s comment. We have deleted this sentence.

Please see the revision in lines 157-158 on page 6 in the revised manuscript.

  1. I doubt that the grains are spherical. They appear polygonal in section and polyhedral in space. Moreover, “grew up” should be revised as “grew”.

Response: Thanks for this comment. We agree with the reviewer that the solid grains are not strictly spherical, we have revised “spheroidized” as “nearly spheroidized”. “grew up” has been revised as “grew”. The revised contents are shown as follows.

Under the isothermal temperature condition of 625 ˚C, the semi-solid material of 6063 aluminum alloy was prepared with the fine, nearly spheroidized, uniform solid grains

Please see line 162 on page 6 in the revised manuscript.

Comparing Figures 2(c) and (d), the solid grains gradually grew and nearly spheridized, and a thicker liquid film was observed separating these solid grains as the isothermal temperature increased from 630 to 635 ˚C.

Please see line 164 on page 6 in the revised manuscript.

  1. “of the microstructures” has been deleted.

Response: “of the microstructures” has been deleted in the paper. The revised sentence is shown as follows.

The characteristics of the microstructure can be obtained by the quantitative analysis of samples processed at different isothermal temperatures.

Please see line 171 on page 6 in the revised manuscript.

  1. “all elevated” should be revised as “increased”.

Response: Thanks for the reviewer’s comment. “all elevated” has been revised as “increased”. The revised sentence is shown as follows.

As shown in Table 3, it was found that the average grain size, shape factor, and effective liquid fraction of solid grains increased with the enlargement in the isothermal temperature.

Please see line 174 on page 6 in the revised manuscript.

  1. Rephrase, these are not characteristic parameters.

Response: “characteristic parameters” has been revised as “microstructural characteristics”. The revised contents are shown as follows.

Microstructural characteristics of 6063 alloy microstructure processed by SSIT at different isothermal temperatures.

Please see line 185 on page 7 in the revised manuscript.

The characteristics of the microstructure can be obtained by the quantitative analysis of samples processed at different isothermal temperatures, which includes the average grain size, the shape factor of solid grains, and the effective liquid fraction.

Please see line 170 on page 6 in the revised manuscript.

  1. I am unfamiliar with the term "deformation regularity"

Response: Thanks for the reviewer’s comment. “deformation regularity” has been revised as “deformation pattern”. The revised contents are shown as follows.

True stress-true strain curves and the deformation pattern

Please see line 187 on page 7 in the revised manuscript.

  1. Please rephrase “It was revealed that at a certain true strain, the true stress reduced with the elevation in the deformation temperature.”

Response: Thanks for the reviewer’s comment. This sentence has been revised. The revised sentence is shown as follows.

It was revealed that at the same value of true strain, the true stress reduced when the deformation temperature increased.

Please see lines 190-191 on page 7 in the revised manuscript.

  1. Please rephrase “At the relatively low deformation temperature, the solid fraction in the sample was high while the solid grains in the microstructure contacted each other to form a spatial framework structure, which led to a low possibility of solid grain slip and rotation.”

Response: This sentence has been rephrased as suggested. The revised sentence is shown as follows.

At the relatively low deformation temperature, the solid fraction in the sample was high, and the solid grains in the microstructure contacted each other to form a spatial framework structure. This framework structure hindered the slip and rotation of the solid grains.

Please see lines 193-195 on page 7 in the revised manuscript.

  1. Please rephrase. There is no polymerization in alloys.

Response: Thanks for the reviewer’s comment. “depolymerize” and “depolymerization” have been revised as “disaggregate” and “disaggregation”, respectively. The revised sentences are shown as follows.

This was because there were more solid grain agglomerates in the sample when the deformation temperature was relatively low, which required a great amount of deformation to disaggregate these agglomerates. On the contrary, when the deformation temperature was relatively high, fewer agglomerates of the larger solid grains were generated in the sample, which was helpful for the completion of the disaggregation without the disaggregation process of the agglomerates or with only a small amount of deformation to achieve the stabilization of the true stress

Please see lines 204-207 on page 7 in the revised manuscript.

  1. Deformation pattern?

Response: Thanks for the reviewer’s comment. “deformation regularity” has been revised as “deformation pattern”. The revised contents are shown as follows.

The deformation pattern of the cold radial forged 6063 aluminium alloy processed by SSIC.

Please see line 219 on page 8 in the revised manuscript.

  1. Rephrase, lines 208-212 in the initial manuscript.

Response: Thanks for the reviewer’s comment. We have rephrased the sentence. The revised sentences are shown as follows.

At the beginning of the SSIC, the central isostatic stress of the cylindrical compression sample is large, which will force the liquid phase to be squeezed and flowed from the central area to the peripheral area. However, the spatial framework structure formed by the solid grains would not be destroyed owing to the small deformation degree, and the deformation of the sample can be recovered if the load is removed. Therefore, the elastic deformation of the spatial framework structure takes place in the sample, namely, the true stress is increased when the true train increased.

Please see lines 225-229 on page 8 in the revised manuscript.

  1. Rephrase, lines 214-223 in the initial manuscript.

Response: Thanks for the reviewer’s comment. This content has been revised. The revision is shown as follows.

Stage II: decrease. As the SSIC continues, when the space framework structure reaches a certain degree of deformation, its deformation cannot be restored even if the load is removed. In other words, the plastic deformation is achieved in the sample. At this stage, the spatial framework structure begins to be destroyed and is gradually and completely surrounded by the liquid phase when the deformation degree increased. The slip and partial rotation of the solid grains then occur at this time. The slip among the solid grains is realized by the shearing force, which promotes the occurrence of holes at the boundary of the grains. These holes would expand rapidly along with the liquid phase, resulting in the damage of samples. This sample damage causes the flow stress to drop rapidly, that is, the true stress drops.

Please see lines 235-250 on pages 8 and 9 in the revised manuscript.

  1. Rephrase, lines 229-230 in the initial manuscript.

Response: Thanks for the reviewer’s comment. This sentence has been revised in the paper. The revision is shown as follows.

The deformation resistance is mainly attributed to the slippage and rotation among the solid grains as well as the liquid phase flow.

Please see lines 258-260 on page 9 in the revised manuscript.

  1. Rephrase, lines 234-237 in the initial manuscript.

Response: Thanks for the reviewer’s comment. This content has been rephrased in the revised manuscript. The revised sentences are shown as follows.

which causes the semi-solid material to roll over during the SSIC process, gradually increasing the contact surface between the pressing head and the sample. This in turn increases the friction between the pressing head and the sample.

Please see lines 266-269 on page 9 in the revised manuscript.

  1. Rephrase, in the deformation temperature elevated at a certain strain rate.

Response: Thanks for the reviewer’s comment. The sentence has been revised, which is shown as follows.

It can be seen that both the peak and steady-state stresses of the aluminum alloy 6063 semi-solid material significantly reduced when the deformation temperature increased at a certain strain rate.

Please see line 278 on page 9 in the revised manuscript.

  1. Rephrase, growth.

Response: Thanks for the reviewer’s comment. “growth” has been changed as “increase”. The revised sentence is shown as follows.

This was because, in the initial stage of SSIC, the liquid fraction in the sample augmented with the increase in the deformation temperature,

Please see lines 281 on page 9 in the revised manuscript.

  1. Rephrase, “the reduction in the connection strength” and “the weakness in its spatial framework structure”

Response: This sentence has been rephrased, which is shown as follows.

This resulted in a decrease of the connection strength between the solid grain aggregates. The spatial framework structure can be more easily destroyed, so the peak stress decreased.

Please see lines 283-285 on page 9 in the revised manuscript.

  1. Rephrase, “enhancement”

Response: “enhancement” has been changed as “increase”. The revised sentence is shown as follows.

It was found that the peak stress of the semi-solid material increased significantly with the increase in the strain rate at a certain deformation temperature.

Please see line 298 on page 10 in the revised manuscript.

  1. Rephrase, “Nevertheless, an opposite change trend in the steady-state stress of the semi-solid material was observed with the augmentation in the strain rate. A reduction tendency was observed in the steady-state stress as the strain rate rose”

Response: Thanks for the reviewer’s comment. We have rephrased the sentence. The revised sentence is shown as follows. 

Nevertheless, a reduction tendency was observed in the steady-state stress with the increase in the strain rate.

Please see lines 299-302 on page 10 in the revised manuscript.

  1. Why and how are the grains damaged?

Response: Thanks for the reviewer’s comment. This sentence has been revised for a better expression, which is shown as follows. 

At the beginning of the SSIC deformation process, the deformation speed of the sample was small because the strain rate was low, which provided enough time for the liquid phase to participate in repairing the liquid phase film among the deformed solid grains.

Please see line 306 on page 10 in the revised manuscript.

On the contrary, the liquid phase did not have enough time to repair the liquid film among the deformed solid grains when the strain rate was high, which caused the difficulty of solid grains to slip and rotate.

Please see line 310 on page 10 in the revised manuscript.

  1. Rephrase, “sample can be well-coordinated with the slippage and rotation among the solid grains”

Response: This sentence has been revised, which is shown as follows. 

The deformation of the spatial framework structure of the sample can take place with the slippage and rotation among the solid grains

Please see line 307 on page 10 in the revised manuscript.

  1. Rephrase, “depolymerization”

Response: “depolymerization” has been revised as “disaggregation”. The revised sentence is shown as follows.

Consequently, it was difficult to achieve the disaggregation and destruction of solid grain aggregates.

Please see line 321 on page 11 in the revised manuscript.

Oppositely, when the strain rate was high, the strong shearing effect between the solid grains can promote the disaggregation and destruction of some solid grain aggregates

Please see line 324 on page 11 in the revised manuscript.

  1. Rephrase, “existed”

Response: Thanks for the reviewer’s comment. This sentence has been revised. The revised sentence is shown as follows.

Semi-solid forming is carried out in the semi-solid temperature range of the metal, where both liquid and solid phases exist.

Please see lines 334-335 on page 11 in the revised manuscript.

  1. Rephrase, “concise formula”

Response: Thanks for the reviewer’s comment. “concise formula” has been changed as “concision”. The revised sentence is shown as follows.

The Arrhenius equation can not only characterize the sensitivity of flow stress to strain rate and deformation temperature but also has the advantages of concision and a high degree of agreement with actual deformation

Please see line 345 on page 11 in the revised manuscript.

  1. Deleted, “power”

Response: “power” has been deleted in the revised manuscript. The revised sentence is shown as follows.

In general, the exponential law expressed in Equation (6) is suitable for the stresses low-level values while the exponential law expressed by Equation (7) is applicable for stresses at high-level values

Please see line 346 on page 11 in the revised manuscript.

  1. Deleted, “recedes”; Rephrase, “augments”

Response: Thanks for the reviewer’s comment. “recedes” has been deleted.  “augments” has been changed as “increases”. The revised sentence is shown as follows.

 shows that the tendency of the increase in the flow stress due to the enlargement in the strain rate when the liquid phase fraction increases.

Please see lines 359-360 on page 12 in the revised manuscript.

  1. Rephrase, “Through”

Response: Thanks for the reviewer’s comment. “Through” has been revised as “After”. The revised sentence is shown as follows.

After calculating the mean value of slopes,

Please see line 367 on page 12 in the revised manuscript.

  1. Rephrase, “linearly”

Response: Thanks for the reviewer’s comment. “linearly fitting method” has been revised as “linear fitting method”.

Please see line 377 on page 12 in the revised manuscript.

  1. Rephrase, “overdetermined equation” and “issue”

Response: Thanks for the reviewer’s comment. This sentence has been revised, which is shown as follows.

The least-squares solution was used to solve the overdetermined equation in this study.

Please see lines 385-386 on page 13 in the revised manuscript.

  1. Rephrase, redundancy “The correlation coefficient between the predicted value of the peak stress by the constitutive equation of the aluminium alloy 6063 in the semi-solid region and the experimentally tested value is shown in Figure 9. It was found that the correlation coefficient in this study reached 0.9796.”

Response: Thanks for the reviewer’s comment. This content has been revised, which is shown as follows. 

As shown in Figure 9, the correlation coefficient between the predicted value of the peak stress and the experimentally tested value was found as 0.9796.

Please see lines 409-412 on page 14 in the revised manuscript.

  1. Rephrase, “Macromorphology results and analysis”, “morphology”

Response: Thanks for the reviewer’s comment. “Macromorphology results and analysis” has been revised as “Macroscopic results and analysis”. “morphology” has been revised as “photo”. The revised contents are shown as follows.

3.4.1. Macroscopic results and analysis

Please see line 420 on page 14 in the revised manuscript.

The macroscopic photo of the radially forged aluminium alloy 6063 semi-solid isothermal compression samples is shown in Figure 10. Figure 10(a) presents the macroscopic photo of the samples processed at the strain rate of 0.1 s-1 with different deformation temperatures.

Please see lines 421-423 on page 14 in the revised manuscript.

Figure 10. Macroscopic photo of samples after the SSIC at different (a) deformation temperatures and (b) strain rates.

Please see line 437 on page 15 in the revised manuscript.

The macroscopic photo of the samples processed at the deformation temperature of 630 ˚C under different strain rates is shown in Figure 10(b). It was found that the macroscopic photo of the compressed sample changed significantly when the strain rate increased.

Please see lines 439-441 on page 15 in the revised manuscript.

The macroscopic photo of the section view along the longitudinal direction of the sample processed by the SSIC deformation is presented in Figure 11(b)

Please see line 449 on page 15 in the revised manuscript.

Figure 11. Deformation characteristics of the compressed samples: (a) the deformation region division and (b) the longitudinal macroscopic photo

Please see line 457 on page 16 in the revised manuscript.

  1. Rephrase, “augmentation”, “was”

Response: Thanks for the reviewer’s comment. “augmentation” has been changed as “increase”. “was” has been changed as “were”. The revised sentences are shown as follows. 

A large increase in the liquid fraction was found in the free deformation region (Region C) compared to the huge deformation region (Region B). The most serious liquid segregates were observed in the free deformation region (Region C).

Please see lines 463-465 on page 16 in the revised manuscript.

  1. Rephrase, “which has a good isolation effect on the solid grains”

Response: This sentence has been rephrased. The revised sentence is shown as follows. 

The liquid phase of the semi-solid material was mainly distributed at the boundary of the solid grains, which has the effect of isolating solid grains.

Please see line 477 on page 17 in the revised manuscript.

  1. Rephrase, “on”

Response: Thanks for the reviewer’s comment. “on” has been revised as “in”. The revised sentence is shown as follows. 

If the deformation continues, severe plastic deformation would take place in the solid grains.

Please see line 481 on page 17 in the revised manuscript.

  1. Rephrase, “spheroidization”

Response: Thanks for the reviewer’s comment. “spheroidization” has been revised as “nearly spherical phenomenon”. The revised sentence is shown as follows. 

The nearly spherical phenomenon was observed in solid grains in the free deformation region (see Positions 4 and 5)

Please see line 497 on page 17 in the revised manuscript.

  1. Rephrase, “difficult and huge deformation regions flew”

Response: This sentence has been revised, which is shown as follows. 

This was mainly because the liquid phase from the hard and huge deformation regions flew into the free deformation region

Please see lines 499-500 on page 18 in the revised manuscript.

  1. Rephrase, “the largest liquid fraction can be observed in the liquid phase flow mechanism”

Response: Thanks for the reviewer’s comment. This sentence has been revised, which is shown as follows. 

As presented in Figure 13(a), the highest liquid phase fraction can be obtained in the liquid phase flow mechanism.

Please see line 520 on page 18 in the revised manuscript.

  1. Rephrase, “the solid rains inside the solid phase grain cluster would slide relative to the adjacent solid grains due to the shear force”

Response: Thanks for the reviewer’s comment. This sentence has been corrected in the revised paper, which is shown as follows. 

The solid grains inside the solid grain cluster would slide relative to the adjacent solid grains due to the shear force

Please see line 534 on page 19 in the revised manuscript.

Reviewer 3 Report

The reviewed work is interesting and constitutes an interesting contribution to the knowledge of the phenomena occurring during the deformation of the material in the area of ​​solid and liquid phases. The first part, which is a literature review, does not require any corrections. However, the work requires a few corrections.

  1. Line 96. How were the liquidus and solidus temperatures determined? The method of their determination and the number of measurements are not presented in the paper. Accurate knowledge of these temperatures is very important due to the nature of the deformation process.
  2. Line 98. No information as to the specific chemical composition of the alloy.
  3. Line 106. Did I understand correctly that asbestos was the insulation material for the stove? Are there no safer insulation materials?
  4. Line 124. How was the deformation value determined?
  5. Lines 159-169. What could be the reason for the observed increase in the value of the shape factor?
  6. Line 172. Have you tried to correlate the obtained shares of the liquid phase from the image analysis with theoretical calculations from the diagram phase?
  7. Line 401. It is worth marking n in the description of Figure 10, what was the strain rate (10a) and what was the temperature (10b).
  8. Line 420. In figure 11a, mark the area of ​​occurrence of zones (A, B, C) so that they comply with 11b.
  9. Lines 524-531. The presented description is unnecessary due to the earlier description of the process.

The mechanisms and phenomena occurring during the deformation of the alloy presented in the paper are convincingly described. However, there is no discussion of the subject of recrystallization of the solid phase and its impact on the final grain size. Does this aspect, according to the authors, not have to be taken into account? There is no information as to what practical application the authors see for the obtained results and the algorithm.

Author Response

The authors would like to take this opportunity to sincerely thank the reviewer for his/her valuable comments. We have further revised the manuscript according to the reviewer's comments. We hope the revised manuscript will meet with your approval. The detailed responses to the reviewer’s comments are provided below.

Comments and Suggestions for Authors

The reviewed work is interesting and constitutes an interesting contribution to the knowledge of the phenomena occurring during the deformation of the material in the area of ​​solid and liquid phases. The first part, which is a literature review, does not require any corrections. However, the work requires a few corrections.

  1. Line 96. How were the liquidus and solidus temperatures determined? The method of their determination and the number of measurements are not presented in the paper. Accurate knowledge of these temperatures is very important due to the nature of the deformation process.

Response: Thanks for the reviewer’s comment. The method for determining the liquidus and solidus temperatures has been added in the revised manuscript. The revised contents are shown as follows.

The solidus and liquidus temperatures of this alloy were determined by the heat flow-temperature curve, which was obtained by using an ‘STA 449F5’ differential scanning calorimeter (DSC). Samples with the dimension of 6mm×6mm×5mm were heated from room temperature to 750°C at 10°C/min under nitrogen atmosphere, and analyzed by DSC. The results show that the solidus and liquidus temperatures of this alloy are 615°C and 655°C, respectively.

Please see lines 98-102 on page 3 in the revised manuscript.

  1. Line 98. No information as to the specific chemical composition of the alloy.

Response: Thanks for pointing this out. The expression of the specific chemical composition has been added in the revised manuscript. Moreover, Table 1 has been corrected for the 6063 aluminum alloy. The revised contents are shown as follows.

The largest weight proportion of the chemical composition is Mg, which accounts for 0.51% and is followed by Si and Mn. Ti and Cr are less than 0.001% and 0.005% in weight percentage, respectively.

Table 1. Chemical composition of the 6063 aluminium alloy (wt.%).

Mg

Si

Mn

Zn

Fe

Cu

Ti

Cr

Al

0.51

0.39

0.03

0.02

0.15

0.01

0.001

0.005

Bal.

Please see lines 102-105 on page 3 in the revised manuscript.

  1. Line 106. Did I understand correctly that asbestos was the insulation material for the stove? Are there no safer insulation materials?

Response: Thanks for the reviewer’s comment. Asbestos was used for the insulation in the experiment. However, it brings the safety issue which we ignored. Therefore, in order to avoid misleading readers, “asbestos” has been changed to “thermal insulation material”. Moreover, we will change the insulation material in our future research for safety. The revised contents are shown as follows.

The SSIC process including SSIT and the deformation of isothermal compression were carried out with the SSIC- equipment which mainly included the top pressing head, bottom pressing head, thermal insulation material, resistance furnace, and thermocouples.

Please see line 115 on page 3 and Figure 1 on page 4 in the revised manuscript.

  1. Line 124. How was the deformation value determined?

Response: Thanks for the reviewer’s comment. The method for determining the deformation value has been added in the revised manuscript. The revised content is shown as follows.

 The reduction in the area is calculated by the ratio of (A0-A1) to A0, where A0 and A1 are the cross-sectional area of the initial alloy and the deformed alloy, respectively.

Please see line 109-110 on page 3 in the revised manuscript.

  1. Lines 159-169. What could be the reason for the observed increase in the value of the shape factor?

Response: Thanks for the reviewer’s comment. The explanation for the increase in the shape factor has been added in the revised manuscript. The revised content is shown as follows.

During the SSIT process, the protruding edges and corners of solid grains will dissolve and subsequently precipitate at the sunken regions due to the difference in curvatures of different parts of the single solid grain [21]. Therefore, the shape factor further improves when the isothermal temperature increases from 630℃ to 635℃.

Please see lines 175-178 on page 6 in the revised manuscript.

  1. Line 172. Have you tried to correlate the obtained shares of the liquid phase from the image analysis with theoretical calculations from the diagram phase?

Response: Thanks for this comment. The correlation of the liquid phase shares from the image analysis with the diagram phase calculation is not carried out in this paper because the image analysis reveals the real liquid phase shares based on experimental results and the diagram phase calculation is not the research content of our study.

  1. Line 401. It is worth marking n in the description of Figure 10, what was the strain rate (10a) and what was the temperature (10b).

Response: Thanks for the reviewer’s comment. The strain rate and the temperature have been added in Figures 10 (a) and (b), respectively. The revised contents are shown as follows.

Please see Figure 10 on page 15 in the revised manuscript.

  1. Line 420. In figure 11a, mark the area of ​​occurrence of zones (A, B, C) so that they comply with 11b.

Response: Thanks for pointing this out. The expression of Zone A, Zone B, and Zone C has been added in Figure 11(a) as suggested. The revised picture is shown as follows.

Please see Figure 11(a) on page 16 in the revised manuscript.

  1. Lines 524-531. The presented description is unnecessary due to the earlier description of the process.

Response: Thanks for the reviewer’s comment. The description of the process has been deleted in the conclusion section.

Please see lines 562-568 on page 19 in the revised manuscript.

  1. The mechanisms and phenomena occurring during the deformation of the alloy presented in the paper are convincingly described. However, there is no discussion of the subject of recrystallization of the solid phase and its impact on the final grain size. Does this aspect, according to the authors, not have to be taken into account? There is no information as to what practical application the authors see for the obtained results and the algorithm.

Response: Thanks for the reviewer’s comment. We agree with the reviewer that the recrystallization would affect the final grain size. However, this paper focus on the effects of the operating parameters on the final microstructure. These operating parameters also affect the nucleation and growth of recrystallized grains, which is finally reflected in the grain size of the microstructure. Therefore, the effect of recrystallization on the final grain size is not discussed in this paper. 

In terms of the practical application of the algorithm obtained from the paper, the constitutive equation established in this paper is used for the calculation of the peak stress and this peak stress is an important factor for selecting the forming equipment of the semi-solid metal forming process. This explanation has been added in the revised manuscript. The added content is shown as follows.

This means that the constitutive equation can accurately express the change in the peak stress of aluminium alloy 6063 in the semi-solid region, which can be used for selecting the forming equipment of the semi-solid metal forming process.

Please see lines 414-415 on page 14 in the revised manuscript.